# Paraphrasing Is All You Need
# for Novel Object Captioning

**Cheng-Fu Yang**[1]       **Yao-Hung Hubert Tsai**[2]       **Wan-Cyuan Fan**[3]

**Ruslan Salakhutdinov**[2]       **Louis-Philippe Morency**[2]       **Yu-Chiang Frank Wang**[3,4]

[1]UCLA       [2]Carnegie Mellon University       [3]National Taiwan University       [4]NVIDIA

cfyang@cs.ucla.edu

## Abstract

Novel object captioning (NOC) aims to describe images containing objects without observing their ground truth captions during training. Due to the absence of caption annotation, captioning models cannot be directly optimized via sequence-to-sequence training or CIDEr optimization. As a result, we present *Paraphrasing-to-Captioning (P2C)*, a two-stage learning framework for NOC, which would heuristically optimize the output captions via paraphrasing. With P2C, the captioning model first learns paraphrasing from a language model pre-trained on text-only corpus, allowing expansion of the word bank for improving linguistic fluency. To further enforce the output caption sufficiently describing the visual content of the input image, we perform self-paraphrasing for the captioning model with fidelity and adequacy objectives introduced. Since no ground truth captions are available for novel object images during training, our P2C leverages cross-modality (image-text) association modules to ensure the above caption characteristics can be properly preserved. In the experiments, we not only show that our P2C achieves state-of-the-art performances on nocaps and COCO Caption datasets, we also verify the effectiveness and flexibility of our learning framework by replacing language and cross-modality association models for NOC. Implementation details and code are available in the supplementary materials.

## 1   Introduction

Novel Object Captioning (NOC) [1] requires one to accurately describe images containing novel objects unseen during training captions. Despite impressive benchmark performance on COCO Captions [2] and Flickr30K [3], existing image captioning models [4, 5, 6, 7, 8] or unsupervised image captioning works [9, 10] cannot generalize well. This is because existing unsupervised captioning models typically assume that training image and caption data share the same visual content (i.e., objects) of interest, which might not be held in practice [9, 10].

Without observing captions describing novel objects during training, a number of NOC works choose to rely on object detection results for filling in the generated slotted sentences [11, 12]. Since the object words and the template sentences are generated separately, their relationships might not be well described. Therefore, Hu et al. [13] propose to relate images and the produced captions by aligning the region features of an object and its associated word embedding, aiming at improved description of novel objects in scenes which are similar to the training ones. However, if a word is not presented in the training corpus, it might not be properly presented in the predicted caption. That is, the word embedding related to novel objects such as verbs and adjectives might not be fully exploited

during inference, resulting in unsatisfactory *linguistic fluency* of output captions. On the other hand, despite the visual feature of a novel object is aligned with textual feature, existing methods generally are not designed to sufficiently caption images containing such objects of interest. In other words, the *fidelity* and *adequacy* of the output captions cannot be preserved.

In this paper, we uniquely approach the task of NOC by introducing and learning paraphrasing capabilities into state-of-the-art captioning models. More specifically, we advance pre-trained language models to expand the word bank of a captioning model for NOC, followed by enforcing the self-paraphrasing ability of this NOC model. The goal of the former stage is to preserve the linguistic fluency for NOC models, while that for the latter stage is deployed to exhibit improved fidelity and adequacy of the learned model. Since no ground truth captions of novel object images are available during training, we apply cross-modality association model with objectives/critics particularly designed for NOC.

It is worth noting that, for the evaluation part of this work, we not only show that our method achieves state-of-the-art performances on the nocaps and COCO Caption datasets, we further assess the metrics reflecting the fluency, fidelity, and adequacy of output captions. In addition, via ablation studies, we further confirm the practicality and flexibility of our learning framework, which does not limit to particular language or cross-modality association modules for paraphrasing and image-text alignment.

## 2   Related Work

**Image captioning.** Recent progress of image captioning focuses on different model architectures and learning methods. Huang et al. [4], Wang et al. [5], Guo et al. [6], Cornia et al. [7] design different attention mechanisms for image captioning. Rennie et al. [14], Li et al. [15], Yang et al. [16] adopt reinforcement learning to improve the performance. On the other hand, some researchers consider more challenging settings, such as partially supervised [17, 18] or unpaired image captioning [9, 10]. However, these methods are restricted to the assumption that the unpaired images and captions share the same set of object class, and the number of object class is limited as well, which make them inapplicable to our task.

**Novel object captioning.** Previously, novel object captioning approaches [19, 20, 21] were only tested on a restrictive dataset with only eight novel object classes held out from the COCO dataset. Their extensions to large-scale image data with various novel objects are not sufficiently studied. Recent studies mainly rely on object detection results to improve the performance on novel object captioning. Lu et al. [11], Wu et al. [12] generate slotted caption templates, which are later filled in with visual concepts identified by object detectors. Yao et al. [22] exploit a copying mechanism to assemble words corresponding to object detector predictions to generate captions. Similarly, Constrained Beam Search (CBS) [19] is an architecture-agnostic decoding algorithm that can be exploited during inference to enforce the inclusion of novel object classes in the captions. Instead of explicitly using detection results, Hu et al. [13] and Vo et al. [23] learn the relationship between image and text by aligning object detection tags with their corresponding image region features. Recently, Wang et al. [24] indicate that a desirable caption should comprise properties of fluency, fidelity, and adequacy. Nevertheless, most existing NOC approaches are not designed to handle language expression and cross-modal association with the above properties preserved.

**Large-scale Vision and Language Pre-training (VLP).** Recently, researchers discover that scaling up the sizes of both captioning model and training dataset would be effective to improve the performance on vision and language tasks. Dual-encoder frameworks such as CLIP [25] and ALIGN [26] scale-up contrastive pre-training [27] using 400M and 1.8B image-text pairs for cross-modal alignment. On the other hand, Transformer-based models like [28, 29] not only scale up the training corpus to 5.65M image and caption pairs, but also increase the transformer layers from 12 to 24. More recently, SimVLM [30] and LEMON [31] further explore the *huge* version of Transformer with a total of 32 layers, and scale up the pre-training corpus with 1.8B and 200M image-text pairs, respectively. Despite the dataset scale as an important factor in image captioning, we will demonstrate that our model still performs favorably against current large-scale methods under the same model size (i.e., Transformer *base* version, 12 layers) and with a smaller training caption corpus.

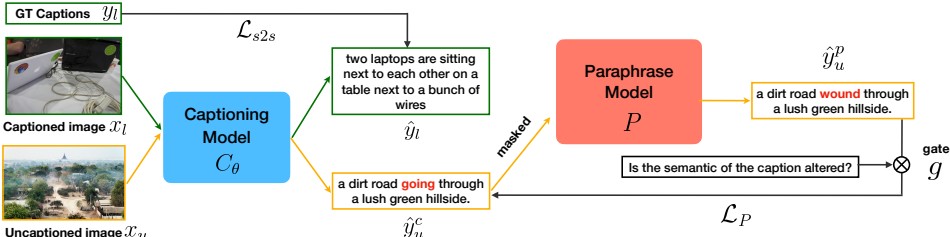

Figure 1: Learning to caption novel objects with linguistic fluency. For caption-labeled image $x_l$, we impose the sequence-to-sequence objective $\mathcal{L}_{s2s}$ for training. For uncaptioned image $x_u$, we exploit $P$ to paraphrase the generated caption $\hat{y}_u^c$, producing the refined caption $\hat{y}_u^p$, followed by a semantic-preserving gate $g$ to verify whether the paraphrased caption has altered the visual content.

## 3 Method

We first determine the notations and settings for the sake of completeness. Given a small set of images $X_l$ with the corresponding captions $Y_l$, as well as a large set of uncaptioned images $X_u$ containing novel objects, our goal is to generate the captions $\hat{Y}_u$ for $X_u$ via a captioning model $C_\theta$ ($\theta$ describes the parameters for the captioning model $C$). To achieve this, we propose *Paraphrasing-to-Captioning (P2C)*, which allows $C$ to perform paraphrasing to preserve linguistic fluency, and to self-paraphrase for boosting the associated fidelity and adequacy. The former is regularized by a pre-trained language model $P$, while an image-text cross-modality model $A$ is utilized for guiding the unsupervised learning process. We note that the contribution of our proposed P2C lies in how to leverage the paraphrasing mechanism, together with the linguistic and visual information from pre-trained models $P$ and $A$, for performing NOC with sufficient caption fluency, fidelity and adequacy.

### 3.1 Describing Novel Objects with Linguistic Fluency

By observing image-caption pairs $(x_l, y_l)$, the captioning model $C_\theta$ in Fig. 1 would learn the visual grounding (i.e., localization of known objects and referring their expressions) as well the linguistics information. The training objective of such labelled data is a conventional sequence-to-sequence loss $\mathcal{L}_{s2s}$, calculated on ground-truth caption $y_l$. That is, we have $\mathcal{L}_{s2s} = \text{CrossEntropy}(\hat{y}_l, y_l)$.

However, using the image-caption pairs $(x_l, y_l)$ alone is not sufficient to produce fluent captions for uncaptioned images $X_u$. We observe that some commonly-used wording of the associated novel objects, are not presented in the training corpus. Therefore, we leverage a language model $P$, to learn its linguistic knowledge via *paraphrasing*. That is, for uncaptioned images $X_u$ containing novel objects, we first generate a caption describing $X_u$. Then, we leverage pre-trained language models as the paraphrase model $P$, which replace the generated captions with the most probable wording of a given novel object context. A semantic-preserving gate $g$ is deployed to validate the paraphrased caption. As detailed later, this gate $g$ is realized by a cross-modal association model $A$, which assesses the relationship between the paraphrased caption and the input image, ensuring semantics of the caption is not modified by $P$.

**Learning to paraphrase via language model $P$.** We now present the detailed paraphrasing processing for learning $C_\theta$. As illustrated in Fig. 1, given an uncaptioned image $x_u$, the captioning model $C_\theta$ generates a caption $\hat{y}_u^c = C_\theta(x_u)$, $\hat{y}_u^c = \{w_1^c, w_2^c, ..., w_T^c\}$, where $w_i^c$ denotes the $i$th word, and $T$ is the caption length. The superscript $c$ represents it is generated by our captioning model. We then obtain a masked caption $\hat{y}_u^M = \{w_1^c, w_2^c, ..., w_m^M, ..., w_T^c\}$ with $m$ indicating the mask index, by randomly masking out the words in the caption. We note that, we do *not* mask nouns/objects in the above process, since they are related to objects grounded in the (novel) visual content and thus are not explicitly associated with caption fluency. As a result, $P$ takes the masked sequence $\hat{y}_u^M$ as input and predicts the masked word $w_m^p$ with the highest probability conditioned on the context of the entire sentence, producing the paraphrased caption $\hat{y}_u^p = P(\hat{y}_u^M)$, $\hat{y}_u^p = \{w_1^c, w_2^c, ..., w_m^p, ..., w_T^c\}$.

**Semantic-preserving gate $g$.** In the above paraphrasing process, however, not every word substitution from $P$ is guaranteed to be semantically correct. We thus require a proper image-text model (i.e., association model) $A$ to validate the paraphrasing output. That is, if the paraphrased caption comprises a more accurate and associated word that human generally uses to describe the scene, then a higher

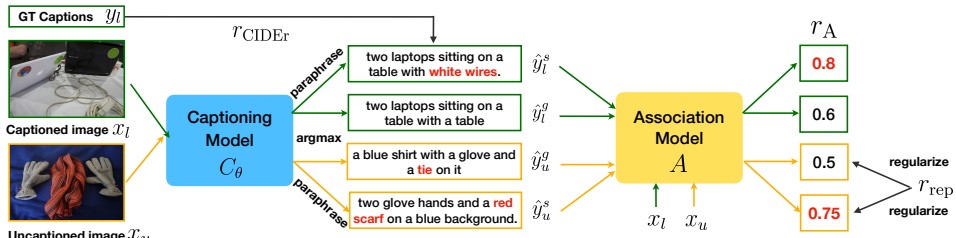

Figure 2: Improving caption fidelity and adequacy via self-paraphrasing. For caption-labeled image $x_l$, we perform CIDEr optimization. The sampled caption $\hat{y}_d^s$ will be rewarded by $A$ if it has a higher cross-modal association than the greedy-decoded baseline $\hat{y}_d^g$. The superscript $d$ indicates the source of the image. Additionally, we regularize our model with $r_{\text{rep}}$ to avoid repetitive captions.

score would be obtained (than that of the original caption). Thus, we propose the objective function $\mathcal{L}_P$ to calculate the loss for the replaced words, gated by the comparison of the association scores of the corresponding captions. More precisely, $\mathcal{L}_P$ is derived as:

$$\mathcal{L}_P = -g * t \log(s), \quad g = \max(0, \tanh(A(x_u, y_u^p) - A(x_u, y_u^c) - \alpha)) \tag{1}$$

where $g$ is the gating function preventing $C_\theta$ from learning from low-quality paraphrased captions, $t$ is the one-hot representation of the paraphrased word, and $s$ is the word distribution predicted by the captioning model at the masked timestep $m$, i.e., we have $w_m^c = \arg\max(s)$. We have $A(x, y)$ in $g$ to calculate the association between an image $x$ and its caption $y$, indicating how well the captions match the images, and $\alpha$ serves as the margin of the association scores for distinguishing between the two captions.

## 3.2 Improving Caption Fidelity and Adequacy via Self-Paraphrasing

Recall that, as discussed in Sec. 1, caption fidelity verifies whether the details of visual content are presented in the generated caption, while adequacy assesses whether the visual content is properly described been expressed in it. While the previous paraphrasing stage increases the linguistic fluency for captions describing images containing novel objects, the novel object word itself still can be incorrectly predicted or missing in the output captions. This is because that, the caption corpus $Y_l$ does not contain any novel objects, and thus the novel object words have low probability to be captioned.

To tackle the above issues, one possible solution is to *self-paraphrasing* for $C_\theta$ with captioning evaluation metric as a critic to reward the paraphrased captions. For labeled images $X_l$, we can exploit the CIDEr score [14], encouraging the generated caption to be consistent with that of the human annotated ones in the word level. However, while CIDEr can be easily computed for captioning labeled images $X_l$, it cannot be explicitly calculated for captioning uncaptioned images $X_u$ due to the absence of ground-truth captions. To address this problem, we discover that cross-modal association is an ideal measurement to reflect the fidelity and adequacy for output captions (see remarks in Appendix A). Specifically, we utilize an association model $A$ to compute the association score between images and the generated captions, which serves as a *label-free* critic in our learning framework. Then, when a paraphrased caption is rewarded by a higher association score, our P2C would increase the probability of using that word in the sentence to describe the image. This learning strategy allows producing captions that precisely describe the objects with plentiful visual details.

**Rewarding the generated captions.** As a potential challenge in NOC, we observe that captioning models would achieve improved association by simply repeating the same object that occurs in the image, which undermines the linguistic fluency of the captions. For example, the image caption *"a group of cans of soda and other items on a table"* can be replaced by *"a pile of **cans** and bottles of soda on a counter with **cans** of **cans**"* with a higher association score.

To tackle this problem, we choose to impose a repetition penalty to avoid such trivial solutions. For image-caption pairs $(x_l, y_l)$, we directly calculate the CIDEr reward for the predicted caption $\hat{y}_l$ (i.e., $r_{\text{CIDEr}} = \text{CIDEr}(\hat{y}_l, y_l)$). To enforce the generated captions for $(X, Y) = (X_l, Y_l) \cup (X_u, Y_u)$ with sufficient fidelity and adequacy, we exploit $A$ to compute the association reward between $X$ and $\hat{Y}$ (i.e., $r_A = A(x, \hat{y})$). As for the repetition penalty to preserve linguistic fluency of the generated captions $\hat{y}_u = \{w1, w2, ..., w_T\}$ for $x_u$, we formulate it as a linear assignment problem, where every

word is assigned to the most similar one in the same sentence except for itself. Then, we calculate the similarity between such pairs for each sentence. Intuitively, repetitive captions would have high similarity scores (with repeating words assigned to the exact same word)s. Thus, we define the assignment $\hat{\alpha}$ as the one that maximizes the average pairwise similarity of a sentence, i.e.,

$$\hat{\alpha} = \arg\max_{\alpha} \frac{1}{T} \sum_{i=1}^{T} C(w_i, w_{\alpha(i)}), \tag{2}$$

where $\alpha(i)$ is the index of the word assigned to the $i$-th word in the caption, and $C(w_i, w_j)$ is the cosine similarity between the GloVe [32] word representation of two words. Since a desirable captioning model would encourage captions with low repetition (i.e., low average pairwise similarity), the reward for repetition penalty is defined as follows:

$$r_{\text{rep}} = 1 - \frac{1}{T} \sum_{i=1}^{T} C(w_i, w_{\hat{\alpha}(i)}), \tag{3}$$

Note that we do not calculate repetition penalty for the labeled data $\hat{Y}_l$ since they are regularized by the aforementioned CIDEr rewards.

With the above discussions, the total reward for caption-labeled data would be $r(\hat{y}_l) = r_{\text{CIDEr}}(\hat{y}_l, y_l) + r_{\text{A}}(x_l, \hat{y}_l)$, and the total reward for uncaptioned data would be $r(\hat{y}_u) = r_{\text{A}}(x_u, \hat{y}_u) + r_{\text{rep}}(\hat{y}_u)$.

**Back-propagation via reinforce algorithm.** Unfortunately, computation of the aforementioned rewards is non-differentiable. Thus, we adopt reinforce algorithm [33] to optimize our P2C learning. As shown in Fig. 2, for an image $x$ we use greedy decoding to obtain the baseline result $\hat{y}^g$, and the paraphrased captions $\hat{y}^s$ are derived from randomly sampling from the word distribution. If the sampled captions possess higher linguistic fluency or cross-modal association than the baseline caption, they will be encouraged by positive rewards and vice versa. We follow Rennie et al. [14], Liu et al. [34, 17] and define the objective function as follows:

$$\nabla_\theta \mathcal{L}_{RL}(\theta) \approx -(r(\hat{y}_d^s) - r(\hat{y}_d^g)) \nabla_\theta \log p_\theta(\hat{y}_d^s),$$
$$r(\hat{y}_d) = \begin{cases} r_{\text{CIDEr}}(\hat{y}_d, y_d) + r_{\text{A}}(x_d, \hat{y}_d) & \text{if } x_d \in X_l \\ r_{\text{A}}(x_d, \hat{y}_d) + r_{\text{rep}}(\hat{y}_d) & \text{if } x_d \in X_u \end{cases}, \tag{4}$$

where $d$ indicates the source of the image, $\theta$ being the parameters of captioning model, and $p_\theta(\hat{y}^s)$ represents the predicted word logits for the generated captions. With the objective functions defined in equations (1), and (4), the NOC model $C_\theta$ can be trained accordingly.

## 4   Experiments

**Implementation details.** Following Hu et al. [13], Li et al. [28], Zhang et al. [29], we consider a BERT-base [35] architecture for our captioning model $C_\theta$. To demonstrate the flexibility of our learning framework, we apply two different versions of BERT, *base* and *large*, which are pre-trained on large-scale *text*-only corpus to model human language as our paraphrase model $P$. For the association model $A$, we have three different cross-modal association models, VIFIDEL [36], SR-PL [17], and CLIP [25] with its version being ViT/B-32. VIFIDEL associates image-caption data using word embedding of particular object labels, and SR-PL utilizes the triplet loss to learn the association of image captions, while CLIP is optimized via the contrastive pre-training. For our model reported in the following subsections, we use BERT large for $P$, and CLIP for $A$. Due to page limits, hyperparameters and other training details can be found in Appendix B.

**Datasets.** The training data for the **nocaps** benchmark comprises the Open Images V4 [37] object detection training set (1.7M images annotated with bounding boxes for 600 object classes), plus the image-caption pairs from the COCO Captions 2017 [2] training set (0.5M image-caption pairs containing 80 object classes). No additional image-caption pairs are provided for training. We evaluate our model on the validation and test set of nocaps, which comprises 4500 and 10600 images from the Open Images validation and test sets, respectively. To compare with current large-scale methods and investigate our model performance when scaling up the training dataset, we jointly trained our model on the Conceptual Caption dataset [38] and compare our method to other methods that access additional image-text pairs during their training on the nocaps-XD [1] benchmark (XD

Table 1: Quantitative results on nocaps. Note that C and S denote CIDEr and SPICE, respectively. We highlight the highest score in blue, while the second best scores are marked in bold.

| Method | in-domain | | near-domain | | out-domain | | overall | | in-domain | | near-domain | | out-domain | | overall | |
|---|---|---|---|---|---|---|---|---|---|---|---|---|---|---|---|---|
| | C | S | C | S | C | S | C | S | C | S | C | S | C | S | C | S |
| | Validation Set | | | | | | | | Test Set | | | | | | | |
| UpDown | 79.3 | 12.4 | 73.8 | 11.4 | 71.7 | 9.9 | 74.3 | 11.2 | 76.0 | 11.8 | 74.2 | 11.5 | 66.7 | 9.7 | 73.1 | 11.2 |
| Oscar$_B$ | 83.4 | 12.0 | 81.6 | 12.0 | 77.6 | 10.6 | 81.1 | 11.7 | 81.3 | 11.9 | 79.6 | 11.9 | 73.6 | 10.6 | 78.8 | 11.7 |
| Oscar$_L$ | 85.4 | 11.9 | 84.0 | 11.7 | 80.3 | 10.0 | 83.4 | 11.4 | 84.8 | 12.1 | 82.1 | 11.5 | 73.8 | 9.7 | 80.9 | 11.3 |
| Oscar$_B$ +VIVO | 92.2 | 12.9 | 87.8 | 12.6 | 87.5 | 11.5 | 88.3 | 12.4 | 89.0 | 12.9 | 87.8 | 12.6 | 80.1 | 11.1 | 86.6 | 12.4 |
| VinVL | **96.8** | 13.5 | 90.7 | 13.1 | 87.4 | 11.6 | 90.9 | 12.8 | 93.8 | 13.3 | 89.0 | 12.7 | 66.1 | 10.9 | 85.5 | 12.5 |
| VinVL +VIVO | 94.8 | 13.3 | **91.4** | 13.0 | 88.7 | 11.6 | **91.4** | 12.7 | **94.5** | 13.1 | **90.9** | 12.9 | 77.9 | 11.3 | **88.3** | 12.6 |
| Human | 84.8 | **14.3** | 85.0 | **14.3** | 95.7 | 14.0 | 87.1 | **14.2** | 80.6 | **15.0** | 84.6 | **14.7** | 91.6 | **14.2** | 85.3 | **14.6** |
| Ours | 101.4 | 15.1 | 96.8 | 14.5 | 95.4 | 12.9 | 97.2 | 14.2 | 101.7 | 15.0 | 95.7 | 14.4 | 82.5 | 12.2 | 93.5 | 14.1 |

stands for extra data). In addition, to demonstrate that our method also enhances model performance on the general image captioning, we evaluate our P2C on the COCO Captions dataset. Due to page limits, the comparison on COCO Captions can be found in Appendix C.2.

## 4.1 Evaluation metrics

**CIDEr.** Like the evaluation metrics [39, 40, 41] in NLP, Consensus-based Image Description Evaluation (CIDEr) calculates the similarity between the reference and generated caption by n-gram overlap in a rule-based manner. To capture human consensus in image captioning, it introduces the tf-idf weight to reduce the matching weight of the n-grams that are common in all image captions.

**SPICE.** Semantic Propositional Image Caption Evaluation (SPICE) [42] matches the semantics between sentences, such as objects, relations, and attributes of objects. Specifically, it converts sentences into semantic scene graphs, which allows evaluation to break grammatical constraints and focuses on propositional semantic content. It reflects the accuracy of the visual content and considers less about linguistic properties.

**Fluency.** To quantitatively evaluate fluency, we remove the effect of the visual information and focus on the quality of linguistic properties in the conventional caption evaluation metrics. Specifically, we we disregard the n-grams containing the particular object word of interest during the computation BLEU@4 [39] and CIDEr scores. Take the sequence "a b c d" for example, when 'b' is the object word, only the unigram "a, c, d" and the bigram "cd" would be taken into account, n-grams such as "ab" or "abc" would be excluded from computation. Note that the fluency experiment is conducted on a subset of the nocaps validation set, which contains 1000 images whose caption annotations are available on the official website of the nocaps dataset.

**Fidelity & Adequacy.** Fidelity and adequacy evaluate how well the captions are associated with images. As defined in Sec. 1, fidelity evaluates whether the objects described by the caption are actually presented in the images, while adequacy assesses how many objects in the images are described by the captions. These two properties are analogous to the definition of precision and recall, respectively. Therefore, we extract the objects mentioned in the captions and the ground-truth objects in the images and calculate the precision (for fidelity), recall (for adequacy), and F1 (for overall association) scores. The experiment is performed on the validation set of nocaps.

Following Agrawal et al. [1], we split the dataset into three subsets for evaluation: *in-domain* images only contain the seen objects that have been described during training, *out-of-domain* images are the unseen/uncaptioned (i.e., novel) ones, while *near-domain* ones contain both seen and unseen objects.

## 4.2 Quantitative analysis

For performance comparisons, we choose UpDown [1] as baselines, as well as Oscar [28] and VinVL [29] which achieves SOTA results on the benchmark of nocaps. All methods are trained via the SCST optimization [14] except for the UpDown baseline, and Constrained Beam Search (CBS) is exploited during inference. In addition, VIVO [13] is a pre-training technique for captioning models, allowing them to recognize the novel objects. Since VinVL did not report the numbers with CBS exploited during inference (CBS is known to improve model performance on out-of-domain data), we reproduce VinVL following details stated in the original paper. For more details please refer to Appendix B. For a comprehensive comparison, we conduct experiments on the validation and test set of nocaps. In addition, we compare our method with VinVL and VIVO in fluency, fidelity,

Table 2: Quantitative results on the nocaps (XD) benchmark. Note that XD stands for extra data.

| Method | Pre-training data | Validation set | | Test set | |
|---|---|---|---|---|---|
| | | CIDEr | SPICE | CIDEr | SPICE |
| Encoder-Decoder [43] | CC12M [43] | 87.4 | 11.8 | 85.3 | 11.8 |
| Encoder-Decoder | CC3M+CC12M | 90.2 | 12.1 | 87.3 | 12.0 |
| VinVL$_{base}$ [29] | 5.65M Combined | 95.5 | 13.5 | - | - |
| SimVLM$_{base}$ [30] | 1.8B | - | - | **94.8** | 13.1 |
| LEMON$_{base}$ [31] | CC3M [38] | 91.6 | 13.0 | - | - |
| LEMON$_{base}$ | CC12M | 100.4 | 13.8 | - | - |
| LEMON$_{base}$ | ALT200M [31] | **106.8** | 14.1 | - | - |
| Ours | COCO Caption 0.5M | 97.2 | **14.2** | 93.5 | **14.1** |
| Ours | CC3M | **104.1** | **14.6** | **102.4** | **14.7** |

Table 3: Quantitative comparisons on caption fluency, fidelity and adequacy. Note that BLEU@4 (B@4) and CIDEr (C) are utilized for describing fluency, object precision (P) for fidelity, object recall (R) for adequacy and object F1 scores (F1) for overall cross-modal association.

| Method | in-domain | | | | | near-domain | | | | | out-of-domain | | | | |
|---|---|---|---|---|---|---|---|---|---|---|---|---|---|---|---|
| | B@4 | C | P | R | F1 | B@4 | C | P | R | F1 | B@4 | C | P | R | F1 |
| VinVL | 21.6 | 74.8 | **59.2** | 40.8 | 48.3 | 19.6 | 73.3 | 22.8 | 32.6 | 26.8 | 17.9 | 59.6 | 48.4 | 25.6 | 33.5 |
| VinVL+VIVO | 20.6 | 71.6 | 56.0 | 42.2 | 48.1 | 19.8 | 75.4 | 28.5 | 36.3 | 32.0 | 17.4 | 59.6 | 49.0 | 27.3 | 35.1 |
| **Ours** | **25.4** | **87.0** | 58.2 | **45.6** | **51.3** | **22.1** | **80.1** | **39.9** | **41.0** | **40.4** | **19.7** | **67.7** | **51.3** | **30.5** | **38.3** |

and adequacy to demonstrate the improvement in terms of these properties. We also evaluate on the COCO Caption dataset and report in Appendix C.2.

**The nocaps datasets.** The results on nocaps are shown in Table 1. From this table, we see that our model performed favorably against baselines and SOTAs across different metrics. We see that our method substantially improves the CIDEr scores in every domain, which verifies our design to generate more fluent and natural captions. Also, it is worth noting that our model largely increased the performance in SPICE score for every data domain, which verifies that our method is able to generate captions with the improved image-language association.

**The nocaps-XD datasets.** To compare with large-scale pre-training methods and investigate our model performance when scaling up training dataset, we evaluate our method on the nocaps-XD benchmark and show the results in Table 2. Note that both VinVL, SimVLM, LEMON, and our method adopt BERT-based backbones for captioning. From the above table, we see that our method still performs favorably against SOTAs on the benchmark even if we have the fewest training caption samples (3M). Our model performance is only slightly below LEMON on the CIDEr score on validation set when it uses a larger training corpus (ALT200M), while we surpass LEMON by a significant margin when the same training corpus (CC3M) is used. Thus, the effectiveness of our method when we scale up the training data can be verified. For more detailed discussion and the ablation study on this model, please refer to Appendix C.3.

**Fluency, fidelity, and adequacy.** As described in Sec. 4.1, we design additional experiments for evaluating fluency, fidelity, and adequacy and report the results in Table 3. For fluency, we remove all the objects and nouns in the captions since they relate less to the linguistics of the captions. We then calculate the BLEU@4 (B@4) and CIDEr (C) scores for the captions after removal. For fidelity and adequacy, they indicate that captions should accurately (high precision) describe sufficient (high recall) visual details. Therefore, we report the object precision and recall in this table, and object F1 scores represent the overall association between captions and images. One can see that our method surpasses previous methods by a visible margin on all tasks except for in-domain object precision, which further verifies our model improves novel object captioning on fluency, fidelity, and adequacy.

### 4.3 Ablation studies

Following the same evaluation procedures in Sec. 4.1, we discuss the contributions of the uses of paraphrasing model $P$ and association model $A$ in terms of linguistic and semantic level metrics, and present their results in Table 4 and 5. In addition, to verify the flexibility of our proposed P2C, we replace $P$ and $A$ with different implementations of language models and association models. Then, we evaluate their performance on the nocaps validation set and report the results in Table 6. Detailed ablation analysis of every objective can be further found in Appendix C.1.

Table 4: Analyses on paraphrasing model $P$, association model $A$, and repetition penalty for NOC using nocaps validation set. Note that $P$ mainly benefits the linguistic fluency with improved CIDEr, and reward from $A$ is desirable for preserving visual semantics with increased SPICE.

| Method | in-domain | | near-domain | | out-domain | | overall | |
|---|---|---|---|---|---|---|---|---|
| | CIDEr | SPICE | CIDEr | SPICE | CIDEr | SPICE | CIDEr | SPICE |
| **Ours** | **102.8** | **14.8** | **97.9** | **14.4** | **86.3** | **12.5** | **96.3** | **14.1** |
| Ours w/o $g$ | 32.8 | 10.6 | 21.5 | 9.5 | 12.6 | 8.0 | 21.3 | 9.4 |
| Ours w/o $\mathcal{L}_P$ | 99.1 | 14.4 | 94.7 | 14.1 | 84.5 | 12.4 | 93.3 | 13.8 |
| Ours w/o $r_A$ | 101.1 | 13.8 | 94.1 | 13.4 | 80.5 | 11.9 | 92.3 | 13.1 |
| Ours w/o $r_{rep}$ | 96.7 | **14.8** | 89.6 | 14.1 | 81.9 | 12.4 | 89.1 | 13.9 |

Table 5: Analyses on paraphrase model $P$ and association model $A$ for improving caption fluency, fidelity and adequacy. Note that $P$ benefits fluency metrics of BLEU@4 and CIDEr, while $A$ focusing on cross-modal association boosts metrics of object precision, recall, and F1 scores.

| Method | in-domain | | | | | near-domain | | | | | out-of-domain | | | | |
|---|---|---|---|---|---|---|---|---|---|---|---|---|---|---|---|
| | B@4 | C | P | R | F1 | B@4 | C | P | R | F1 | B@4 | C | P | R | F1 |
| **Ours** | **25.4** | **87.0** | 58.2 | **45.6** | **51.3** | **22.1** | **80.1** | 39.9 | **41.0** | **40.4** | **19.7** | **67.7** | 51.3 | **30.5** | **38.3** |
| Ours w/o $\mathcal{L}_P$ | 21.6 | 77.1 | 58.1 | 40.9 | 48 | 19.8 | 75.5 | **41** | 39.0 | 40.0 | 19.1 | 66.1 | **51.6** | 19.1 | 66.1 |
| Ours w/o $r_A$ | 23.0 | 79.3 | **58.8** | 42.1 | 49.1 | 21.2 | 78.6 | 35.8 | 37.7 | 36.8 | 18.8 | 65.7 | **51.6** | 27.4 | 35.8 |

**Paraphrase model $P$.** As shown in Table 4, the captioning model without $P$ would observe a performance drop in CIDEr for linguistic fluency, but such drops for the metric of SPICE (related to visual content) would be less significant. Similarly, as observed in Table 5, removing $P$ would result in the lowest BLEU and CIDEr scores. These results confirm our motivation and model design, since $P$ is utilized to improve caption quality at the linguistics level.

To further verify the use and flexibility of paraphrase model $P$, we replace our paraphrase model $P$ (the *large* version of BERT) with the *base* version of BERT. We found out that only a slight performance drop is produced. We conjecture that this is because the two versions of BERT are trained on the same text corpus, which means they share the same word bank. As a result, our model would distill similar linguistic knowledge when using either model to guide the training of our P2C.

**Semantic-preserving gate $g$.** From Table 4, we see that captioning model would be misled by the wrong guidance produced by $P$, if no validation from $g$ to ensure that the semantics of the original captions is not modified by $P$. This results in a significant performance drop on nocaps.

**Association model $A$.** As shown in Table 4, the model without the reward calculated by $A$ showed significant drops in captioning metrics of CIDEr and SPICE. However, the performance decrease in SPICE is expected, since $A$ is particularly deployed for preserving visual content in captions. As for CIDEr, its decrease is mainly due to the deterioration of missing visual content in captions. This is also confirmed by Table 5, in which mainly the metrics reflecting fidelity and adequacy would observe significant drops for model trained without the association rewards.

To further verify the use and flexibility of the association model $A$, we replace it with different models that also produce association scores between images and captions. Specifically, we consider VIFIDEL [36], SR-PL [17], and CLIP [25]. As the results shown in Table 6, one can see that the use of CLIP as $A$ would achieve the best performance. This is because that, VIFIDEL only associates image-caption data using word embedding of particular object labels, while CLIP assesses such cross-modal data in the instance level, i.e., taking the complete caption of an image into consideration. As for SR-PL, it utilizes the triplet loss to learn the association of images and captions, while CLIP is optimized via the contrastive pre-training, which fully exploits the negative samples in a mini-batch to learn a more compact representation space, allowing it to estimate the association more accurately.

**Repetition penalty.** Recall that, in Sec. 3.2, this penalty is to alleviate the association between images and captions with redundant visual information. As seen in Table 4, the model without this penalty observed a significant performance drop in CIDEr scores. We did not see such trends for SPICE. This is because that, repetitive words in captions mainly violate linguistic structures rather then semantic accuracy, and thus the performance related to linguistic fluency would be more sensitive to the deployment of this penalty.

Table 6: Uses of different paraphrase models $P$ and association models $A$ for NOC. Results on the nocaps validation set are presented. Note that Stages 1 and 2 are the paraphrasing schemes presented in Sec. 3.1 and Sec. 3.2, respectively.

| Method | Paraphrase Model $P$ | Association Model $A$ | $\mathcal{L}_P$ | $r_A$ | $r_{rep}$ | Overall CIDEr | SPICE |
|---|---|---|---|---|---|---|---|
| Baseline | N/A | N/A | ✗ | ✗ | ✗ | 81.2 | 12.3 |
| Ours w/ Stage 1 | BERT$_{base}$ | CLIP [25] | ✓ | ✗ | ✗ | 84.1 | 12.7 |
| Ours w/ Stage 1 | BERT$_{large}$ | CLIP | ✓ | ✗ | ✗ | 84.2 | 12.7 |
| Ours w/ Stage 2 | BERT$_{large}$ | VIFIDEL [36] | ✓ | ✓ | ✓ | 55.3 | 10.7 |
| Ours w/ Stage 2 | BERT$_{large}$ | SR-PL [17] | ✓ | ✓ | ✓ | 84.6 | 13.2 |
| Ours w/ Stage 2 | BERT$_{large}$ | CLIP | ✓ | ✓ | ✓ | **96.3** | **14.1** |

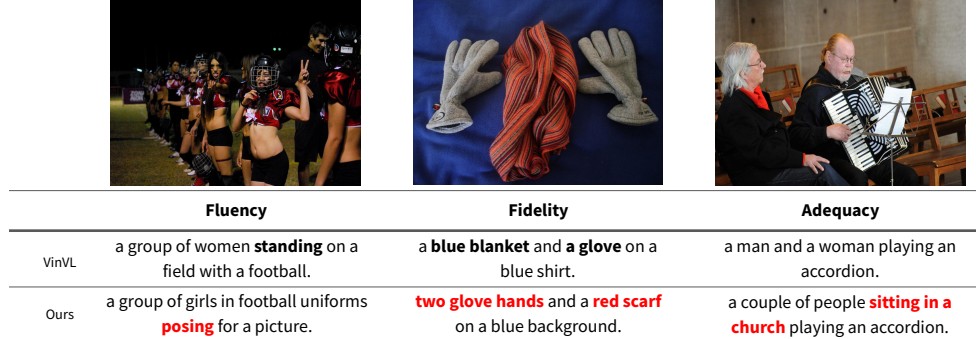

| | Fluency | Fidelity | Adequacy |
|---|---|---|---|
| VinVL | a group of women **standing** on a field with a football. | a **blue blanket** and **a glove** on a blue shirt. | a man and a woman playing an accordion. |
| Ours | a group of girls in football uniforms **posing** for a picture. | **two glove hands** and a **red scarf** on a blue background. | a couple of people **sitting in a church** playing an accordion. |

Figure 3: Example results and comparisons for image captions produced by VinVL and ours in terms of fluency, fidelity and adequacy. Note that both utilize VIVO for novel object detection.

## 4.4 Qualitative analysis

For qualitative comparisons, we first conduct human study and ask individuals to evaluate a caption from three perspectives: fluency, fidelity, and adequacy. Due to page limit, the experiment details and the corresponding results are presented in Appendix C.4. For qualitative analysis, we empirically show captions in Fig. 3, which are generated by our model and VinVL, with both pre-trained from VIVO for novel object detection. In this figure, wordings that are less accurate or incorrectly describe the associated visual content are marked in bold. And, our wording improvements are highlighted in red. From this figure, one can see that for fluency, our model generated vivid captions with more proper wordings. Take the image on the left for example, our model particularly described *"posing for a picture"* instead of *"standing on a field"*. As for fidelity, our model is designed to accurately capture the visual content in an image. Specifically, take the second image for example, we correctly described the number of gloves and the novel object *red scarf*, while VinVL failed to do so. As for adequacy, take the bottom-right image for example, our model was able to recover visual details in the image (i.e., *"people playing the accordion"* and *"sitting in a church"*). For more qualitative examples, please refer to Appendix C.5.

## 5 Conclusion

In this paper, we proposed Paraphrasing-to-Captioning (P2C) for novel object captioning, with particular goals of improving caption fluency, fidelity, and adequacy. In P2C, we advocate the learning of two paraphrasing capabilities for captioning models. The first is language-level paraphrasing, which expands the word bank of a captioning model under the guidance of pre-trained language model, and thus preserves the linguistic fluency during NOC. Secondly, we introduce the self-paraphrasing ability for the captioning model to sufficiently describe visual content of the input image, so that both caption fidelity and adequacy can be achieved. Due to the lack of ground truth captions, image-text association is uniquely exploited for guiding the training process. Empirically, we not only showed that our model achieved SOTA results on benchmark datasets, we also assessed the metrics associated with fluency, fidelity, and adequacy, confirming the effectiveness of our model. Via our ablation studies, we further verified the flexibility of our learning framework by replacing language and cross-modality association modules for paraphrasing and image-text alignment. Finally, since we

apply pre-trained language and association models in P2C, their joint optimization would be among our future research directions.

**Acknowledgement** This work is supported in part by the Ministry of Science and Technology of Taiwan under grant NSTC 110-2634-F-002-052. Y.-H.H.T., R.S., and L.-P.M. was supported in part by the NSF IIS1763562, NSF Awards #1750439 #1722822, National Institutes of Health, IARPA D17PC00340, ONR Grant N000141812861, and Facebook PhD Fellowship. We also thank to National Center for High-performance Computing (NCHC) for providing computational and storage resources.

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
