# OpenReview forum: "Paraphrasing Is All You Need for Novel Object Captioning"
_NeurIPS.cc/2022/Conference — NeurIPS 2022 Accept_

### Official Review · Reviewer_B4GQ · 2022-07-05

**Rating:** 7
**Confidence:** 3
**Soundness:** 4 excellent
**Presentation:** 3 good
**Contribution:** 3 good

**Summary:**

## Summary
This paper performs Novel Object Captioning (NOC), which is to describe images containing novel objects that were not seen during the training phase. The data consists of both captioned and un-captioned images. For captioned images, a Captioning model $C_{\theta}$ generates a caption which is compared against the ground truth caption. For un-captioned images, $C_{\theta}$ first generates a caption, \hat{y}\_u^c, and is masked at a certain word index to produce on \hat{y}\_u^m, which is then fed into the paraphrase model, $P$, to perform masked language modeling to predict the masked word. A semantic-preserving gate, $g$ is used to validate \hat{y}\_u^m for semantic correctness. $g$ consists of an association model, $A$ which calculates the association between an image and a caption. The caption corpus might not contain words that can be used to describe novel objects, so to improve caption fidelity and adequacy, self-paraphrasing. When the association model, $A$ provides a higher score to a caption, $P2C$ would increase the probability of using that word to describe the image. To avoid repetition, the words in a sentence are paired with all other words and the similarity of a sentence is calculated. Sentences with high repetition would have a high similarity score. A repetition score $r_{rep}$ is calculated using the above similarity scores. Since $r_{rep}$ is not differentiable, a reinforce algorithm is used to reward captions that have higher linguistic fluency or cross modal association than a baseline caption obtained from greedy decoding. Their model beats the existing methods by outperforming on metrics like CIDEr, SPICEm Fluency, Fidelity and Adequacy and achieves results similar to human evaluations. Through ablation studies, the importance of each component in the model is shown.

**Questions:**

## Questions
1. Is the captioning model first trained on captioned images before moving on to un-captioned images?
2. Some details about which pre-trained language models were used as the paraphrase model $P$ would be appreciated.
3. How many times is masking performed on a single generated caption, $\hat{y}_u^c$? Do you mask out different indexes and provide each masked caption as a unique item to the model?

**Limitations:**

## Limitations
The authors have mentioned that the core components of their model relies on pre-trained models for performing paraphrasing and image to text association and have not performed a joint optimization of these models.

**Strengths And Weaknesses:**

## Strengths
The authors have identified that the captioning corpus for captioned images does not contain important words that can be used to describe novel objects, thus a lot of the existing methods fail. They overcome this limitation by formulating the problem as a paraphrasing task. They also show effectiveness of existing language and association models as core components of their model.

## Weaknesses
One major limitation is that the paraphrasing is highly dependent on the initial caption generated by the captioning model and not much attention has been given on this step. If the initial generated caption performs poorly, the subsequent steps might not produce a high quality caption.

---

> ### Author Response · Authors · 2022-08-02
> **Response to Reviewer B4GQ**
>
> We thank Reviewer B4GQ for the positive comments and suggestive remarks. Please see our responses below for each raised issue.
>
> **Q1**: "One major limitation is that the paraphrasing is highly dependent on the initial caption generated by the captioning model and not much attention has been given on this step. Is the captioning model first trained on captioned images before moving on to un-captioned images?"
>
> **A1**: We thank the reviewer for pointing out this practical issue. We are aware that NOC performance would be sensitive to model initialization. As a result, following previous NOC works like VinVL and VIVO, we first train our captioning model using pairwise image-caption data (COCO Captions) with 20 epochs, then we will move on to joint training of captioned and uncaptioned images (the algorithm described in Sec. 3.1).
>
> **Q2**: "Some details about which pre-trained language models were used as the paraphrase model  would be appreciated."
>
> **A2**: We thank the reviewer for giving us the opportunity to clarify this issue. As noted in L188 of the original manuscript, we choose BERT base and BERT large as our paraphrase model. We also provide ablation studies in Table 4 to analyze the effect of different choices of the paraphrase model.
>
> **Q3**: "How many times is masking performed on a single generated caption, y^c_u? Do you mask out different indexes and provide each masked caption as a unique item to the model?"
>
> **A3**: We thank the reviewer for giving us the opportunity to clarify this issue. For each generated caption $\hat{y}^c_u$, we would randomly mask out two words in the caption at the same time (L579 at stage1.py in our supplementary materials) to produce the masked caption $\hat{y}^p_u$, and provide such caption to our paraphrase model. Since our model is trained using this algorithm with 20 epochs, a significant portion (if not all) of the indexes will be masked out during the training stage. Therefore, we do not generate multiple masked captions for a single generated caption  $\hat{y}^c_u$ in each training iteration. We will update this information in L558 in the supplementary materials.

---

### Official Review · Reviewer_HSxb · 2022-07-11

**Rating:** 7
**Confidence:** 3
**Soundness:** 3 good
**Presentation:** 2 fair
**Contribution:** 3 good

**Summary:**

In image captioning, generating expressions that refer to the objects
that have never appeared in the training data is one of the important
issues.  To improve fluency as well as fidelity and adequacy in
generating captions, this paper introduces a two-stage training
process with a support of techniques related to automatic paraphrase
generation in each stage.  In the first stage of training, the
generated raw caption is randomly masked and fed into a BERT model to
predict the most probable word for it as a paraphrase, and if the
paraphrased caption has a positive effect according to a cross-modal
association model, the cross-entropy loss between the raw and
paraphrased captions is back-propagated to update the parameters.  In
the second stage of training, the sampled output is regarded as a
paraphrase of the greedily decoded caption, and their quality gap is
measured by the cross-modal association model so that the difference
is used to update the parameters in the reinforcement learning
fashion.  An empirical experiment on the "nocaps" dataset confirms
that the proposed method can generate better captions than existing
methods in terms of two automatic evaluation metrics: CIDEr and SPICE.
The ablation studies using the validation set also shows that the
components of the proposed method function as the authors intended.

**Questions:**

I have several questions regarding the experiments.

- Among two options for $P$ and three options for $A$, what are chosen?

- What do the blue and bold numbers in Table 1 stand for?

- When computing the fluency scores, the authors "remove" all the
objects and nouns from the captions.  Did they truly "remove" them?
For instance, removing "b" from "a b c d" we can no longer distinguish
the adjacency of "a c" and "c d" which looks problematic.  Removal
also leads to shorter sequences, and consequently the significant
reduction of the total number of longer n-grams will make the scores
less reliable.

- What does "-" in Table 2 indicate?  If the authors copied some
numbers from previous publications, they are not comparable with the
scores computed by the authors themselves, due to the difference of
the computational environments, toolkits, etc.

**Limitations:**

No.  The proposed method has a limitation: it assumes two large pre-trained models and cannot be applied to languages for which such prerequisites are not available.
The proposed method exploits neural generative models but their usage is limited to labeling given pairs of sentences.  Therefore, the trained models will not introduce any societal influence nor misleading conclusions.

**Strengths And Weaknesses:**

I acknowledge the following two strengths of this paper.

- The proposed method can seek a wider search space for potential
captions, by utilizing pre-trained language model and sampling-based
decoding.  As a result, it can generate captions of higher fluency,
fidelity, and adequacy, compared to existing methods.

- The paper reports on ablation studies to confirm the contribution of
each of newly introduced components.  It justifies the proposed
approach with regard to the motivation, and enables readers to better
understand the proposed method.

The paper contains several flaws and has some readability issues that
hinder readers from understanding the proposed method correctly and
easily, as well as hurts the credibility of the paper.

- There are a couple of notational errors for the proposed method.
For instance, in Section 3.1, $w^c_i$ denotes the $i$-th word
(l.113).  To introduce the masked caption, the masked token is noted
as $w^m_M$ and $m$ is introduced as the mask index.  However,
$w^c_M$ should be $w^c_M$ and $m$ should be $M$.  In Eq.(1),
the cross entropy is calculated between two words (surface forms),
which is impossible and thus it should be corrected.  In l.160, the
set of inputs, $X$, and another set of outputs, $\hat{Y}$, are
defined.  However, they should be derived as
$(X,\hat Y)=(X_l,\hat Y_l)\cup(X_u,\hat Y_u)$; otherwise,
the pairs in the two component sets will be lost.

- The paper is not self-contained.  I appreciate that the authors made
an enormous effort especially for conducting various experiments.
However, as a result of claiming too many things in the main paper
without describing even minimum explanation for them, the readability
and clarity are hurt.  I recommend to prune some statements so that
the main paper is self-contained and reader-friendly.

- The descriptions in Section 4 are sometimes loose.  Line charts in
Figures 3 and 4 are inappropriate because they show the results for
nominal scales: the distance between in and near, and that for near
and out are not the same.  I ask some questions below.

---

> ### Author Response · Authors · 2022-08-02
> **Response to Reviewer HSxb (Part 1 of 4)**
>
> We thank Reviewer HSxb for the positive comments and suggestive remarks. Please see our responses below for each raised issue.
>
>
> **Q1**: "The paper contains several flaws and has some readability issues that hinder readers from understanding. For example, there are a couple of notational errors for the proposed method. In addition, the paper is not self-contained. I recommend pruning some statements so that the main paper is self-contained and reader-friendly."
>
> **A1**: We sincerely thank the reviewer for providing careful and detailed reviews. Please see our revised version below, and we will improve the readability in our final version.
> **a**. Changing the notation of the masked token from $w^m_M$ to $w^M_m$ with the subscript $m$ as the mask index and the superscript $M$ denoting the masked word.
> **b**. Changing the cross-entropy equation (Eq. (1)) to $L_p = - t log(s)$, where $t$ is the one-hot representation of the paraphrased word $w^p_m$, and $s$ is the word distribution predicted by the captioning model $C$ at the masked timestep $m$, i.e., we have $w^c_m = \operatorname{argmax}(s)$.
> **c**. Changing the notation of the image-caption pairs from $X=X_l ∪ X_u$, $Y = Y_l ∪ Y_u$ to $(X,Y) = (X_l, Y_l) ∪ (X_u, Y_u)$ to keep the image-caption pairwise characteristics.

---

> > ### Author Response · Authors · 2022-08-02
> > **Response to Reviewer HSxb (Part 2 of 4)**
> >
> > **Q2**: "The descriptions in Section 4 are sometimes loose. Line charts in Figures 3 and 4 are inappropriate because they show the results for nominal scales: the distance between in and near, and that for near and out are not the same."
> >
> > **A2**: We understand the concern, and we thank the reviewer for the suggestive comment. As originally noted in L262, Figure 3 is used to quantitatively compare our method with SoTA methods using the metric of fluency, fidelity and adequacy, while Figure 4 is designed to ablate the paraphrase model P and the association model A using the same metrics as Fig. 3. In these figures, we present the results from different domains to demonstrate that the performance trend is consistent across all of the domains. For example, one can clearly see that our model surpasses previous methods in all five metrics and all of the domains, and in Figure 4 we observe that removing the paraphrase model hurts the fluency metrics the most (the red line), while the removal of the association model affects fidelity and adequacy the most (the green line).
> >
> > Nevertheless, we understand the concern from the reviewer that readers may get confused, since the nominal scales of captioning performance would vary with different caption domains. Therefore, we choose to replace Figure 3 and 4 with the tables below (Table A & B) to improve the readability of the paper. We thank the reviewer again for helping us strengthen our work.
> >
> > Table A. Quantitative comparisons on fluency, fidelity, and adequacy. Note that both BLEU@4 (B@4) and CIDEr (C) measure caption fluency, object precision (P), recall (R) and F1 scores (F1) assess fidelity, adequacy and visual-linguistic association, respectively.
> > |             | in-domain |      |      |      |      | near-domain |      |      |      |      | out-domain |      |      |      |      |
> > |:----------- |:----------:|:----:|:----:|:----:|:----:|:-----------:|:----:|:----:|:----:|:----:|:----------:|:----:|:----:|:----:|:----:|
> > |             |    B@4    |  C   |  P   |  R   |  F1  |    B@4     |  C   |  P   |  R   |  F1  |    B@4    |  C   |  P   |  R   |  F1  |
> > | VinVL       |    32.6    | 63.2 | **59.2** | 40.8 | 48.3 |    30.5     | 58.3 | 22.8 | 32.6 | 26.8 |    29.6    | 48.8 | 48.4 | 25.6 | 33.5 |
> > | VinVL +VIVO |    31.2    | 59.8 | 56.0 | 42.2 | 48.1 |    30.1     | 57.3 | 28.5 | 36.3 | 32.0 |    27.3    | 45.6 | 49.0 | 27.3 | 35.1 |
> > | Ours        |    **35.9**    | **68.0** | 58.2 | **45.6** | **51.3** |    **32.2**     | **60.8** | **39.9** | **41.0** | **40.4** |   **30.1**    | **50.2** | **51.3** | **30.5** | **38.3** |
> >
> > Table B. Analyses on P and A for improving caption fluency, fidelity and adequacy. Note that the paraphrase model P benefits fluency metrics (i.e., BLEU@4 (B@4) and CIDEr (C)), while the association model A focusing on visual-linguistic association and thus boosts object precision (P), recall (R), and F1 scores (F1).
> > |                        | in-domain |          |          |          |          | near-domain |          |          |          |          | out-domain |          |          |          |          |
> > |:---------------------- |:----------:|:--------:|:--------:|:--------:|:--------:|:-----------:|:--------:|:--------:|:--------:|:--------:|:----------:|:--------:|:--------:|:--------:|:--------:|
> > |                        |    B@4    |    C     |    P     |    R     |    F1    |    B@4     |    C     |    P     |    R     |    F1    |    B@4    |    C     |    P     |    R     |    F1    |
> > | Ours                   |  **35.9**  | **68.0** |   58.2   | **45.6** | **51.3** |  **32.2**   | **60.8** | **39.9** | **41.0** | **40.4** |  **30.1**  | **50.2** |   51.3   | **30.5** | **38.3** |
> > | Ours w/o L_P |    33.8    |   64.8   |   58.1   |   40.9   |   48.0   |    30.4     |   60.6   |   41.0   |   39.0   |   40.0   |    27.9    |   49.1   | **51.6** |   27.9   |   36.2   |
> > | Ours w/o r_A |    33.2    |   65.9   | **58.8** |   42.1   |   49.1   |    31.9     |   60.7   |   35.8   |   37.7   |   36.8   |    28.7    |   49.8   | **51.6** |   27.4   |   35.8   |
> >
> > **Q3**: "Among two options for P and three options for A, what are chosen?"
> >
> > **A3**: We thank the reviewer for giving us the opportunity to clarify this issue. As noted in the implementation details in Sec. 4 (L188), we choose BERT base and BERT large for P to see how the capacity of language models would affect our model performance. As for the association model A, we have VIFIDEL [36], SR-PL [17], and CLIP [25] with its version being ViT/B-32. VIFIDEL associates image-caption data using word embedding of detected object labels, and SR-PL utilizes the triplet loss to learn the association of image captions, while CLIP is optimized via the contrastive pre-training.

---

> > > ### Author Response · Authors · 2022-08-02
> > > **Response to Reviewer HSxb (Part 3 of 4)**
> > >
> > > **Q4**: "What do the blue and bold numbers in Table 1 stand for?"
> > >
> > > **A4**: We thank the reviewer for pointing this out. We highlight the highest score in blue, while the second best scores are marked in bold. We would update the captions accordingly.
> > >
> > > **Q5**: "When computing the fluency scores, the authors "remove" all the objects and nouns from the captions. Did they truly "remove" them? For instance, removing "b" from "a b c d" we can no longer distinguish the adjacency of "a c" and "c d" which looks problematic. Removal also leads to shorter sequences, and consequently the significant reduction of the total number of longer n-grams will make the scores less reliable."
> > >
> > > **A5**: We thank the reviewer for raising up this practical issue, and we are glad to offer additional clarification. We would like to first point out that existing methods [1, 2] generally use subjective evaluation (i.e., human study) to assess the fluency of predicted captions. In our paper, we also conduct human study and report the results in Table 8 in the supplementary material (see below).
> > >
> > > |            | M1 (Turing Test) | M2 (Fidelity) | M3 (Adequacy) | M4 (Fluency) |
> > > | ---------- |:----------------:|:-------------:|:-------------:|:------------:|
> > > | VinVL+VIVO |       0.25       |     3.70      |     3.46      |     3.99     |=
> > > | Ours       |       0.43       |     4.33      |   **4.24**    |     4.06     |
> > > | Human      |     **0.53**     |   **4.44**    |     4.18      |   **4.09**  |
> > >
> > >
> > > In addition to human study, we aim to provide quantitative evaluation in a more objective way. To achieve this, we actually remove all the object words from both the ground-truth and predicted captions. Take the sequence “a b c d” for example, we simply remove the word ‘b’ from the sequence and use the new sequence “a c d” for the calculation of fluency score. Since the removal is performed on every captions, we believe the comparison on fluency scores would be fair.
> > >
> > > Nevertheless, we understand the particular concern from the reviewer that shortening sentences may lead to less reliable scores. Therefore, we choose an alternative way to calculate the fluency scores while preserving the n-gram characteristics of the caption. Specifically, instead of truly “removing” the object words from the caption, we disregard the n-grams containing the particular object word of interest during the computation of BLEU and CIDEr. Take the previous sequence “a b c d” for example, when ‘b’ is the object word, only the unigram “a, c, d” and the bigram “cd” would be taken into account, n-grams such as “ab” or “abc” would be excluded from computation. In this way, undesirable n-grams such as “ac” will not be taken into account while calculating the fluency metric. We report the updated fluency scores in the table below (Table C&D). From the Table D, we observe the same trends in our original experiment (Fig. 4) that removing the paraphrase model P would produce the lowest BLEU and CIDEr scores. In other words, P is introduced to improve caption quality at the linguistics level can be again verified.
> > >
> > > [1] Fluency-Guided Cross-Lingual Image Captioning. Lan et al., ACM Multimedia 2017.
> > > [2] Informative Image Captioning with External Sources of Information. Zhao et al., ACL 2019.
> > >
> > >
> > > Table C. Quantitative comparisons on caption fluency.
> > > |            | in-domain |        | near-domain |          | out-domain |          |
> > > | ---------- |:---------:|:------:|:-----------:|:--------:|:-------------:|:--------:|
> > > |            |   BLEU    | CIDEr  |    BLEU     |  CIDEr   |     BLEU      |  CIDEr   |
> > > | VinVL      |   21.6    |  74.8  |    19.6     |   73.3   |     17.9      |   59.6   |
> > > | VinVL+VIVO |   20.6    |  71.6  |    19.8     |   75.4   |     17.4      |   59.6   |
> > > | Ours       | **25.4**  | **87** |  **22.1**   | **80.1** |   **19.7**    | **67.7** |
> > >
> > > Table D.  Analyses on the impact of paraphrase model P and association model A on caption fluency.
> > > |                        | in-domain |       | near-domain |       | out-domain |       |
> > > | ---------------------- |:---------:|:-----:|:-----------:|:-----:|:-------------:|:-----:|
> > > |                        |   BLEU    | CIDEr |    BLEU     | CIDEr |     BLEU      | CIDEr |
> > > | Ours                   |   25.4    |  87   |    22.1     | 80.1  |     19.7      | 67.7  |
> > > | Ours w/o L_P |   21.6    | 77.1  |    19.8     | 75.5  |     19.1      | 66.1  |
> > > | Ours w/o r_A |    23     | 79.3  |    21.2     | 78.6  |     18.8      | 65.7  |

---

> > > > ### Author Response · Authors · 2022-08-02
> > > > **Response to Reviewer HSxb (Part 4 of 4)**
> > > >
> > > > **Q6**: "What does "-" in Table 2 indicate? If the authors copied some numbers from previous publications, they are not comparable with the scores computed by the authors themselves, due to the difference of the computational environments, toolkits, etc."
> > > >
> > > > **A6**: We thank the reviewer for giving us the opportunity to clarify this issue. Since we directly copied all the numbers in Table 2 from previous publications (for direct and fair comparisons), we use “-” in Table 2 to represent that previous publications did not report such results. We would also like to point out that, among all the methods presented in Table 2, our method utilizes the fewest image-caption pairs (3M), which is significantly less than other SoTA methods like LEMON (200M) and SimVLM (1.8B). As a result, we would require less computation resources (trained with 8 V100 GPU) when compared to, for example, SimVLM (trained with 512 TPU v3 chips). Thus, we believe the comparisons in Table 2 would still be informative.

---

> > > > > ### Comment · Reviewer_HSxb · 2022-08-08
> > > > > **Thank you for your response.**
> > > > >
> > > > > I appreciate the detailed responses from the authors, but they do not answer to all of my questions.
> > > > >
> > > > > Q3: Among two options for $P$ and three options for $A$, what are chosen?  In their response, they just repeated the options described in ll.186-193.  The models used as $P$ and $A$ for "Ours" in Tables 1, 2, and 3 are still unclear.  By the way, I understood that these different options are compared in the ablation studies (Section 4.3 and Table 4).

---

> > > > > > ### Author Response · Authors · 2022-08-08
> > > > > > **Updated Response to Q3.**
> > > > > >
> > > > > > **Updated A3**: We apologize for the misunderstanding. For P and A used in Tables 1, 2, and 3, BERT_large is selected as our paraphrase model P, and CLIP is utilized as the association model A. We will clarify this information in L194.

---

> > > > > > > ### Comment · Reviewer_HSxb · 2022-08-08
> > > > > > > **Thank you for your confirmation**
> > > > > > >
> > > > > > > I wish the next version of the paper will be much more self-contained.

---

### Official Review · Reviewer_YAnp · 2022-07-12

**Rating:** 7
**Confidence:** 4
**Soundness:** 3 good
**Presentation:** 4 excellent
**Contribution:** 3 good

**Summary:**

This paper proposed an interesting approach for novel object caption generation, where the main idea is to improve the captions via first leveraging a pre-trained language model for paraphrasing to generate more linguistically fluent paraphrases, and then performing self-paraphrasing based on a cross-visual association model. The models are rewarded to generate captions with low repetitions within the framework of the reinforce algorithm to optimize the overall learning process. Experiments and evaluations on nocaps, nocaps (XD), and MSCOCO datasets show the effectiveness of the proposed model.

**Questions:**

N/A

**Limitations:**

See weaknesses above.

**Strengths And Weaknesses:**

Strengths:

- Overall, the proposed idea appears to be solid and backed by a series of thoroughly carried out experiments.

- Both automatic and human evaluation have been carried out as well as rigorous ablation studies were conducted on multiple different datasets. Results show the effectiveness of the proposed components over the considered baselines and SOTA models by a considerable margin.

- The shared codebase would be beneficial for the research community.

- The paper is very well-written and organized to communicate the key ideas and contributions properly.

Weaknesses:

- The proposed P and A models are built on existing pre-trained language and cross-modal association models.

- I would be interested to see how the proposed models perform in generating captions for out-of-domain (e. g. clinical domain) image captioning datasets (e. g. see https://www.imageclef.org/2022/medical/caption).

---

> ### Author Response · Authors · 2022-08-02
> **Response to Reviewer YAnp**
>
> We thank Reviewer YAnp for the positive comments and suggestive remarks. Please see our responses below for each raised issue.
>
> **Q1**: "The proposed P and A models are built on existing pre-trained language and cross-modal association models."
>
> **A1**: We understand the particular concern of using existing pre-trained models in our proposed framework, and we are glad to further clarify this issue. We would like to first point out that, we did not claim the uses of these pre-trained models being the contributions. As noted in Sec. 1, we address the challenging task of novel object captioning (NOC), which requires models to accurately describe images containing novel objects without seeing the ground-truth captions. Without ground-truth supervision, we introduce a novel framework that can heuristically optimize the output captions via paraphrasing, together with pre-trained language and cross-modal association models to provide training signals. Specifically, the pre-trained language model is utilized to expand the word bank of a captioning model for NOC (Sec. 3.1), and the association model serves as a proxy to assess the quality of output captions (Sec. 3.2). Moreover, we also provide fundamental support in Sec. A. to justify our design, explaining how we are able to utilize and relate language model P and association model A, leveraging their intrinsic knowledge for producing captions with fluency, fidelity, and adequacy guarantees.
>
>
> **Q2**: "I would be interested to see how the proposed models perform in generating captions for out-of-domain (e.g., clinical domain) image captioning datasets (e. g. see https://www.imageclef.org/2022/medical/caption)."
>
> **A2**: We thank the reviewer for the interesting yet challenging suggestion. We would like to point out that both NOC and standard image captioning (e.g., COCO Captions) aim to describe objects in general scenes, not for particular domains. Considering that the style or syntax of clinical domain captions would be fundamentally different from captions used in NOC or COCO, we do not expect our model (or existing NOC works) would generate satisfactory captions for such a distinct domain. We will list the above issue as a common limitation shared by NOC works, and list this among the future research directions.

---

> > ### Comment · Reviewer_YAnp · 2022-08-07
> > **Thank you**
> >
> > Thank you authors for the detailed clarification and comments. Looking forward to your future work in this area!

---

### Official Review · Reviewer_azLR · 2022-07-12

**Rating:** 6
**Confidence:** 4
**Soundness:** 3 good
**Presentation:** 3 good
**Contribution:** 3 good

**Summary:**

In this paper, the authors proposed a ‘Paraphrasing-to-Captioning (P2C)’ framework for Novel Object Captioning (NOC), which is an interesting task. Overall, four modules in two stages are proposed to improve the adequacy, fidelity, and fluency of NOC. Specifically, through performing paraphrasing in the first stage, the proposed model is trained to describe novel objects with linguistic fluency. In addition, self-paraphrase is proposed for boosting the associated fidelity and adequacy. Finally, experiments evaluated on the nocaps benchmark are reported. It shows improvements compared to existing methods such as VinVL+VIVO.

**Questions:**

- In Table 2, the reported results of 'VinVL+VIVO' are different from the results reported in [3]. The authors may explain the possible reason behind this.

[3] NOC-REK: Novel Object Captioning with Retrieved Vocabulary from External Knowledge, CVPR 2022

**Limitations:**

Yes, limitations have been discussed. I do not find the potential negative societal impact of this work.

**Strengths And Weaknesses:**

[Strengths]
- This paper is well-written and easy to follow. Basically, its motivation is reasonable and soundness. After reading this paper, I can basically understand why this model can achieve better performance than existing methods, i.e., the experimental comparisons basically validate the usefulness of the proposed methods.

- The NOC task presented in this paper is interesting to me. I am glad to see the authors can utilize uncaptioned images and unpaired texts in their training.

- Fluency, fidelity and adequacy are the three most important components in captioning tasks. The authors employ various strategies to ensure the generated captions have these properties.

[Weakness]
- Important ablation studies are missing. For example, if we hope to justify the effectiveness of paraphrasing, the results of 'Baseline+r_CIDER+r_A' should also be compared.

- I am wondering about the role of the association model in this paper. On the one hand, I am not sure if the comparisons are fair, since the association model employed visual-linguistic pair during their training. On the other hand, employing CLIP provided reward for training captioning models may not be new, [1] and [2] presented similar ideas.

[1] CLIPScore: A Reference-free Evaluation Metric for Image Captioning, emnlp 2021
[2] Fine-grained Image Captioning with CLIP Reward, CVPR 2022

---

> ### Author Response · Authors · 2022-08-02
> **Response to Reviewer azLR (Part 1 of 3)**
>
> We thank Reviewer azLR for the positive comments and suggestive remarks. Please see our responses below for each raised issue.
>
> **Q1**: "Important ablation studies are missing. For example, if we hope to justify the effectiveness of paraphrasing, the results of 'Baseline+r_CIDEr+r_A' should also be compared."
>
> **A1**: We thank the reviewer for giving us the opportunity to clarify this issue.  In our original manuscript, we have conducted ablation studies in Table 3, which confirms that paraphrasing would improve caption fluency.
>
> For the sake of clarity, we present Table 3 below for discussions. The first row in Table 3 is the full version of our proposed model. The full version of our model is equivalent to ‘Baseline+L_P+r_CIDEr+r_A+r_rep’. To verifies the effectiveness of the introduced paraphrasing module, we remove the paraphrase module in our proposed framework and report the numbers in the third row (i.e., Ours w/o L_P). This model is equivalent to 'Baseline+r_CIDEr+r_A+r_rep' as suggested. Comparing the first and third rows across different settings, we see that the model without paraphrasing experienced performance drops on CIDEr, while those on SPICE were less significant. This is because CIDEr reflects caption fluency by measuring the n-gram overlapping between prediction and ground-truth captions, while SPICE is used to evaluate the visual content presented in the captions.
>
> |                     | in-domain |          | near-domain |          | out-domain |          | overall  |          |
> | ------------------- |:---------:|:--------:|:-----------:|:--------:|:----------:|:--------:|:--------:|:--------:|
> |                     |   CIDEr   |  SPICE   |    CIDEr    |  SPICE   |   CIDEr    |  SPICE   |  CIDEr   |  SPICE   |
> | Ours                | **102.8** | **14.8** |  **97.9**   | **14.4** |  **86.3**  | **12.5** | **96.3** | **14.1** |
> | w/o $g$             |   32.8    |   10.6   |    21.5     |   9.5    |    12.6    |   8.0    |   21.3   |   9.4    |
> | w/o L_P   |   99.1    |   14.4   |    94.7     |   14.1   |    84.5    |   12.4   |   93.3   |   13.8   |
> | w/o r_A   |   101.1   |   13.8   |    94.1     |   13.4   |    80.5    |   11.9   |   92.3   |   13.1   |
> | w/o r_rep |   96.7    | **14.8** |    89.6     |   14.1   |    81.9    |   12.4   |   89.1   |   13.9   |
>
>
> In addition to Table 3, we conduct alternative ablation studies in Table 5, in which we sequentially add each proposed module to the proposed framework. We present the table below for the ease of discussions. The first row in Table 5 represents the baseline result, and the second row reports the results after deploying the paraphrase module. From the table below one can see that  our model achieved a notable improvement on the CIDEr metric, which echoes our conclusion that the paraphrase module would benefit the resulting caption fluency.
>
> | Method                             | in-domain  |           | near-domain |           | out-of-domain |           |  overall  |           |
> |:---------------------------------- |:----------:|:---------:|:-----------:|:---------:|:-------------:|:---------:|:---------:|:---------:|
> |                                    |   CIDEr    |   SPICE   |    CIDEr    |   SPICE   |     CIDEr     |   SPICE   |   CIDEr   |   SPICE   |
> | Baseline (Only w/ L_s2s) |   89.07    |   13.29   |    83.29    |   12.61   |     68.77     |   10.59   |   81.17   |   12.32   |
> | +L_P                     |   92.46    |   13.4    |    85.79    |   12.92   |     73.21     |   11.40   |   84.2    |   12.69   |
> | +r_CIDEr                 |   101.19   |   13.84   |    95.38    |   13.44   |     83.24     |   12.06   |   93.75   |   13.23   |
> | +r_A                     |   96.73    | **14.83** |    89.64    |   14.12   |     81.87     |   12.38   |   89.08   |   13.88   |
> | +r_rep                   | **102.77** | **14.83** |  **97.9**   | **14.40** |   **86.33**   | **12.54** | **96.25** | **14.10** |

---

> > ### Author Response · Authors · 2022-08-02
> > **Response to Reviewer azLR (Part 2 of 3)**
> >
> > **Q2**: "I am wondering about the role of the association model in this paper. I am not sure if the comparisons are fair, since the association model employed visual-linguistic pairs during their training.""
> >
> > **A2**: We thank the reviewer for raising the potential issue of unfair comparison. We understand that, the use of pre-trained association models (e.g., CLIP) implies the access of additional visual-linguistic data pairs during training. However, such models only associate image and caption data pairs at the instance level (not word level) during pre-training. Our captioning model has no access to these image-caption pairs nor the ground truth captions containing novel objects during training. We would like to note that, recent NOC works also utilize external knowledge for solving this challenging task. For example, a recent CVPR’22 work [3] requires object descriptions crawled from Wiktionary to learn the concept of novel object for NOC.
> >
> > Nevertheless, we agree and understand the above issue might cause fairness concerns during experiments. This is the reason why, in our paper, we compare our work to other SOTA methods that also use extra data during training. The results are shown in Table 2 (attached below). We note that LEMON utilizes additional 200M caption samples, and SimVLM requires 1.8B captions. As for our work, while the association model CLIP is pre-trained on a 400M image-caption corpus, it only observes instance-level information during pre-training. As for training our captioning model, only 3M image-caption pairs are available. From this table, it can be seen that our model still performed favorably against the above models (with remarkably more visual-linguistic data pairs utilized) with a significant margin.
> >
> > | Method                     | Training data     | Validation set |          | Test set  |          |
> > | -------------------------- | ----------------- |:--------------:|:--------:|:---------:|:--------:|
> > |                            |                   |     CIDEr      |  SPICE   |   CIDEr   |  SPICE   |
> > | Encoder-Decoder [43]       | CC12M [43]        |      87.4      |   11.8   |   85.3    |   11.8   |
> > | Encoder-Decoder            | CC3M+CC12M        |      90.2      |   12.1   |   87.3    |   12.0   |
> > | VinVL_base [29]  | 5.65M Combined    |      95.5      |   13.5   |     -     |    -     |
> > | SimVLM_base [30] | 1.8B              |       -        |    -     |   94.8    |   13.1   |
> > | LEMON_base [31]  | CC3M[38]          |      91.6      |   13.0   |     -     |    -     |
> > | LEMON_base       | CC12M             |     100.4      |   13.8   |     -     |    -     |
> > | LEMON_base       | ALT200M [31]      |   **106.8**    |   14.1   |     -     |    -     |
> > | Ours                       | COCO Caption 0.5M |      97.2      |   14.2   |   93.5    |   14.1   |
> > | Ours                       | CC3M              |     104.1      | **14.6** | **102.4** | **14.7** |
> >
> > **Q3**: "Employing CLIP provided reward for training captioning models may not be new, [1] and [2] presented similar ideas."
> >
> > **A3**: We thank the reviewer for giving us the chance to clarify our contribution. We would like to point out that we did not claim the use of CLIP among our contributions. As noted in Sec. 1, the key contribution of our work is to address novel object captioning without observing the associated ground-truth caption data. We introduce a novel learning framework that can heuristically optimize the output captions via paraphrasing, together with pre-trained language and cross-modal association models to provide training signals. Specifically, the pre-trained language model is utilized to expand the word bank of a captioning model for NOC (Sec. 3.1), while the association model serves as a proxy to assess the quality of output captions (Sec. 3.2). In addition, we also provide fundamental support in Sec. A. of our supplementary materials, explaining why the computed association score can jointly improve the fidelity and adequacy of resulting captions.
> >
> > For the suggested work of CLIPScore [1], it applies CLIP for caption evaluation only (not during training). As for Cho’s work of [2], it addresses the task of general image captioning, not NOC. They require ground-truth captions to calculate the CIDEr reward to avoid the model from overfitting on the CLIP reward. While we uniquely propose a repetition penalty that can be computed on the predicted captions, allowing our framework to generalize well in the absence of ground-truth captions.

---

> > > ### Author Response · Authors · 2022-08-02
> > > **Response to Reviewer azLR (Part 3 of 3)**
> > >
> > > **Q4**: "In Table 2, the reported results of 'VinVL+VIVO' are different from the results reported in [3]. The authors may explain the possible reason behind this."
> > >
> > > **A4**: Since VinVL+VIVO is only reported in Table 1, we assume that the reviewer has concerns on the results of ‘VinVL+VIVO’ in that table. The numbers are different from those in [3], because we applied Constrained Beam Search (CBS) to every model in Table 1 during inference (as we mentioned in L236 of the original manuscript). We utilize this technique because CBS is known to improve model performance, especially on out-of-domain data [4]. From the table below one can clearly see that after applying CBS, the captioning performance significantly improves on the out-of-domain images. Since the original VinVL+VIVO paper does not report numbers that exploit CBS during training and the pre-trained model is not available, we reproduce this method following the instructions in their paper, and we present the implementation details in Sec. B. in our supplementary materials.
> > >
> > >
> > > |                | in-domain |       | near-domain |       | out-domain |       | overall |       |
> > > | -------------- |:---------:|:-----:|:-----------:|:-----:|:----------:|:-----:|:-------:|:-----:|
> > > |                |   CIDEr   | SPICE |    CIDEr    | SPICE |   CIDEr    | SPICE |  CIDEr  | SPICE |
> > > | VinVL+VIVO     |   94.9    |  13.0   |    91.1     | 12.9  |    79.1    | 11.2  |  89.2   | 12.6  |
> > > | VinVL+VIVO+CBS |   94.8    | 13.3  |    91.4     |  13.0   |    88.7    | 11.7  |  91.4   | 12.7  |
> > >
> > > [4] nocaps: novel object captioning at scale. Agrawal et al., ICCV 2019.

---

### Meta-Review · Area_Chair_BRPP · 2022-08-28

**Recommendation:** Accept
**Confidence:** Certain

**Metareview:**

All reviewers appreciated this paper's simple and intuitive ideas on promoting fluency, fidelity and adequacy in novel-object captioning (NOC) task via paraphrasing modules. They also appreciated the good results on multiple benchmarks and also human evaluation, plus the good writing. The authors also have very detailed useful responses in the rebuttal period. Some suggestions made to the authors to incorporate were clearer ablation tables, out-of-domain task discussion, self-contained and more clear notations and formulations.

**Award:**

No

---

### Decision · Program_Chairs · 2022-09-14

Accept